# Evolution of *Streptococcus pneumoniae* Serotype 3 in England and Wales: A Major Vaccine Evader

**DOI:** 10.3390/genes10110845

**Published:** 2019-10-25

**Authors:** Natalie Groves, Carmen L. Sheppard, David Litt, Samuel Rose, Ana Silva, Nina Njoku, Sofia Rodrigues, Zahin Amin-Chowdhury, Nicholas Andrews, Shamez Ladhani, Norman K. Fry

**Affiliations:** 1Vaccine Preventable Bacteria Section, Public Health England–National Infection Service, London NW9 5EQ, UK; Carmen.Sheppard@phe.gov.uk (C.L.S.); david.litt@phe.gov.uk (D.L.); Samuel.rose@phe.gov.uk (S.R.); catariinagas@gmail.com (A.S.); njoku.nina@gmail.com (N.N.); asofiabrodrigues@gmail.com (S.R.); norman.fry@phe.gov.uk (N.K.F.); 2Immunisation and Countermeasures, Public Health England–National Infection Service, London NW9 5EQ, UK; Zahin.Amin@phe.gov.uk (Z.A.-C.); shamez.ladhani@phe.gov.uk (S.L.); 3Statistics, Modelling and Economics, Public Health England–National Infection Service, London NW9 5EQ, UK; nick.andrews@phe.gov.uk

**Keywords:** *Streptococcus pneumoniae*, serotype 3, whole genome sequencing

## Abstract

Despite its inclusion in pneumococcal conjugate vaccine 13 (PCV13), *Streptococcus pneumoniae* serotype 3 remains a major cause of invasive pneumococcal disease in England and Wales. Previous studies have indicated that there are distinct lineages within serotype 3 clonal complex 180 and the clade distributions have shifted in recent years with the emergence of clade II. We undertook whole genome sequencing and genomic analysis of 616 serotype 3 isolates from England and Wales between 2003 and 2018, including invasive and carriage isolates. Our investigations showed that clade II has expanded since 2014 and now represents 50% of serotype 3 invasive pneumococcal disease (IPD) isolates in England and Wales. Genomic analysis of antibiotic resistance and protein antigen genes showed that distinct profiles are present within the clades which could account for the recent emergence of this clade. This investigation highlights the importance and utility of routine whole genome sequencing and its ability to identify new and emerging variation at the single nucleotide level which informs surveillance and will impact future vaccine development.

## 1. Introduction

Pneumococcal conjugate vaccines (PCVs) have been pivotal in reducing the incidence of invasive pneumococcal disease (IPD) despite the emergence of non-vaccine serotypes. However, *Streptococcus pneumoniae* serotype 3 continues to be among the major causes of IPD despite its inclusion in PCV13 and vaccine effectiveness has been reported as non-significant for this serotype [1], leading to it being recorded as a non-vaccine type in some vaccine efficacy studies [2]. The low vaccine efficacy has been linked to the lack of covalent linkage of the capsular polysaccharide to peptidoglycan, resulting in polysaccharide release [3]. Furthermore, when compared to other serotypes associated with IPD, serotype 3 has a high case carrier ratio and children have been shown to carry high levels of antibody to serotype 3, presumed to be due to a high rate of natural exposure but low duration of carriage [2,4,5], supporting the suggestion that this serotype is highly invasive. 

Pneumococcal isolates are delineated by their serotype, determined by the capsular operon or resulting polysaccharide, or sequence type (ST), produced by conventional seven gene multilocus sequence typing (MLST). STs may also be grouped into larger clusters called clonal complexes (CCs), containing related sequence types (single or double locus variants). Clonal complex 180 (CC180) is the major clonal complex associated with serotype 3 and does not offer any discrimination between the majority of serotype 3 isolates. Previous studies have shown that although CC180 appears to contain very closely related isolates, the accessory genome shows high levels of variation and it is possible to split this CC into different clades [6,7]. The study of European isolates by Croucher et al. [6] showed that most of the isolates were within a single clade (clade I); however, two major clades were observed in a global study by Azarian et al. [7], clade I (including subclades Ia and Ib) and clade II. This study used CC180 serotype 3 isolates from various studies across a large time frame (1993–2014). These data suggest a shift in the serotype 3 population and that clade II has emerged in recent years showing a genomic divergence from pre-PCV13 isolates. Clade II is not observed in the early study years (1993–1998) and increases in number after 2005. A further study of carriage isolates from Massachusetts [8] also noted genomic changes in serotype 3 after the introduction of PCV13, despite the overall proportion of this serotype remaining constant. 

The study by Azarian et al. [7] also showed that the different clades presented distinct antigenic and antibiotic resistance profiles, with clade II showing higher levels of antimicrobial resistance than clade Ia. These differences are suggested to be the reason that clade II has begun to emerge in recent years. 

We used available archived isolates and existing whole genome sequence (WGS) data from Public Health England (PHE) IPD surveillance during the years 2003–2019 (*N* = 616) to investigate whether the increase in serotype 3 in the data from England and Wales in recent years was due to an increase in a previously unseen clade and whether this could be the reason for the vaccine evasion of serotype 3. These data provide an important initial analysis of a large dataset from pre- and post-PCV era in a single population.

## 2. Materials and Methods

### 2.1. Selection of Isolates

Routine WGS of invasive pneumococcal isolates was introduced in October 2017 at Public Health England. Therefore, to obtain retrospective isolates for sequencing, the laboratory information management system was queried for all routine referred serotype 3 isolates received from November 2003 (when the archiving systems began) to the end of 2016—excluding strains received for quality control purposes but including non-invasive isolates submitted to the reference laboratory within this period. Due to records beginning in the later part of 2003, all isolates were selected for sequencing. Therefore, if isolates could not be located, did not grow or were incorrectly identified as serotype 3 by original slide agglutination, a replacement was not available for sequencing. 

To create a diverse temporal and geographical distribution of the historical isolates, the remaining isolates were ordered by reception date. Approximately 10% of isolates were sequenced from the total number within each year apart from 2003. If the sending hospital of an isolate selected for sequencing was the same as one selected within the previous calendar month, an alternative isolate was selected for sequencing. In the case of dead, contaminated or missing samples, the next sequential isolate (subject to referring hospital restrictions) was selected as a replacement. 

For isolates from 2017 and 2018, a search identified all isolates that had been sequenced as part of the routine service and the process as described above was used to select 10% of isolates from these lists. The proportion of isolates from 2017 included in the study was 8% as routine sequencing was not in place from January of that year; therefore, to maintain the temporal distribution in other years, fewer isolates were available for selection. In addition to these isolates, 14 serotype 3 carriage isolates from previous sequencing studies [4,9,10,11] were included. In total, 616 isolates were included in the analysis, representing ca. 10% of all serotype 3 isolates submitted to the reference laboratory.

Genomes included in the study have been uploaded to the European Nucleotide Archive (ENA) and accession numbers are provided in Appendix A.

### 2.2. DNA Extraction and Sequencing

DNA extraction and sequencing were carried out as described in Kapatai et al. [12]. The initial bioinformatic analysis included kmer identification to confirm the presence of *S. pneumoniae* (https://github.com/phe-bioinformatics/kmerid), MLST using MOST [13] and genomic serotyping by PneumoCaT [12].

### 2.3. Kmer Cluster Analysis

To identify the relationship between the previous serotype 3 clade analysis and the PHE dataset, WGS data were downloaded from ENA for the genomes in the previous study [7] and analysed together with the PHE data as described below. 

Genomes were assembled using unicycler (version v0.4.7) [14] using the default parameters. The assemblies were then used in PopPUNK (version 1.1.3) [15] with a minimum kmer size of 13 and a maximum number of mixture components (--K) of 3. Output was visualized using cytoscape (version 3.7.1) [16] using the organic layout algorithm [17]. 

### 2.4. Phylogenetic Analysis

The in-house PHE pipeline, PHEnix (https://github.com/phe-bioinformatics/PHEnix), was used to produce a reference-based alignment of WGS data from all PHE isolates (*N* = 616). The *S. pneumoniae* serotype 3 clade Ia isolate OXC141 (NC_017592.1) was used as a reference and isolates were mapped using BWA (version 0.7.12) [18] and variants called using GATK Unified Genotyper (version 2.6.5) [19] with the following options: sample ploidy = 2; genotype likelihoods model = both; rf = BadCigar. The following filters were applied to the VCF files: allelic depth ratio = 0.9; minimum depth = 10; minimum qual score = 40; minimum mapping quality = 30; mapping quality 0 ratio = 0.1. 

To reduce the number of false positives, regions within the genome impacted by mapping issues were identified and excluded. To identify these regions, reads without error were simulated using wgsim (https://github.com/lh3/wgsim) using the following parameters: base error rate = 0; number of read pairs = 3,000,000; read length = 100 bp; rate of mutations = 0; fraction of indels = 0; probability of indel extension = 0. The resulting reads were then subject to mapping and variant calling as described above. A BED file of positions that could not be accurately identified by GATK was produced to be excluded from the alignment. 

Once VCF files were produced, they were translated into a FASTA alignment using the “vcf2fasta” function within PHEnix. Additional filters were applied to reduce the proportion of *N* bases and gaps: columns where the proportion of *N* bases or gaps exceeded 0.5 were removed; samples for which the proportion of *N* bases or gaps was greater than three standard deviations above the mean number of *N* bases or gaps, respectively. For this dataset, the mean number of *N* bases per sample was 0.008147 and the standard deviation was 0.008131; therefore, the threshold for *N* bases per sample was 0.032541 and seven samples were removed from the analysis. The mean number of gaps per sample was 0.021106 and the standard deviation was 0.023525; therefore, the threshold for gaps per sample was 0.091680 and no additional samples were removed. 

The resulting FASTA file contained 488779 bp for comparison. RAxML (version 8.1.17) [20] was used to produce a maximum likelihood tree using the following arguments: algorithm = a (rapid Bootstrap analysis); random number seed = 12345; parsimony random seed = 12345; number of alternative runs = 100; model of substitution = GTRGAMMA. The resulting tree was visualized using ggtree [21].

Tempest [22] was used to identify suitability of the dataset for BEAST analysis. 

### 2.5. Clade Determination

PopPUNK clusters were used to relate PHE genomes to data from previous studies, and clade definitions were assigned to clusters that contained an isolate from the Azarian et al. [7] dataset. Nodes were subsequently identified in the PHEnix phylogeny as defining each of the clades (Ia, Ib, II) to assign PopPUNK singletons and smaller “PHE only” clusters to their respective clades. There were no instances of isolates defined as different clades appearing in the same PopPUNK cluster.

### 2.6. Antigen Variant Detection and Antibiotic Resistance Prediction

The 13 pneumococcal protein antigens assessed in Azarian et al. [7] were also assessed in this dataset using SRST2 [23] with default values. Due to its high variability, additional partial *PspA* sequences from previous studies [24,25] were included in an additional database and the threshold for maximum coverage was reduced to 80% and the maximum divergence to 20% in SRST2.

The WGS pipeline describe in Metcalf et al. [26] was shared with PHE and modified to accept the assemblies generated as described above to identify any antimicrobial resistance (AMR) markers. 

Assemblies generated were annotated using Prokka (version 1.13.7) [27] and the GenBank file for strain OXC141 was preferentially used for annotation. Resulting GFF files were inspected for isolates that showed AMR markers to identify the presence of potential transposons or mobile genetic elements.

### 2.7. Capsular Operon Comparisons

The capsular operons for all PHE isolates were compared using the PHEnix pipeline as described above. Serotype 3 operon sequence CR931634.2 from Bentley et al. [28] was used as a reference. In this case, no columns were removed if the proportion of Ns or gaps was > 0.5 and all samples were retained for comparison. Bcftools [29] was used to combine VCFs from isolates within clades Ia and clade II so they could be compared. The isec command was used to identify any variants that were consistently present or absent in either clade Ia or clade II.

## 3. Results

### 3.1. Isolate Sequencing

Table 1 shows a breakdown of the number of serotype 3 isolates per year. The number of serotype 3 carriage isolates included are noted in a separate column. 

### 3.2. Identification of Clades

PopPUNK analysis was used to compare data from Azarian et al. [7] to the PHE dataset. Figure 1 shows the clusters produced by this analysis. Clusters were assigned to a clade using the original clade definition from the Azarian et al. data [7]. 

The clusters produced by PopPUNK show two major groups for clades Ia and II. It is also apparent that the major clusters for clade II are generally associated with later years. Figure 2 shows the proportion of isolates within each clade broken down by year. The graph also shows the total proportion of serotype 3 disease within the PHE IPD data for that year. These data indicate that the proportion of isolates sequenced is generally representative of the proportion of IPD cases caused by serotype 3 in England and Wales. The emergence of clade II in the years following the introduction of the vaccine in 2010 is clear. These changes have not been shown to be associated with age or gender within this dataset. Figure 3 shows a time trend analysis for the rate per million people of each clade over time. This shows that there is a small decrease in clade Ia (average of 2.3%, *p* = 0.035) and there is no significant change in the rate of external isolates across this period (*p* = 0.81). Clade II shows a significant increase after 2013 (*p* = 0.017) and this was clear by 2015 (*p* ≤ 0.001).

### 3.3. Phylogenetic Analysis

Phylogenetic analysis (Figure 4) showed the expected structure—two major clades (clade I and II), with a larger proportion of the clade I isolates falling within a separate “major” clade (clade Ia). Interestingly, there is a group of isolates within CC505 that falls within clade Ib. ST 505 (46,8,2,10,6,1,22) is a double locus variant of ST180 (7,15,2,10,6,1,22), and thus it is not unexpected that these isolates may have a similar genetic background. All other clonal complexes were outside the recognized clades and are referred to as “external”.

Within clade Ia, the two major clades appear to separate into distinct branches due to consistent variation. However, within clade II, this does not seem to be the case and the two major clusters are mixed within the clade. As indicated by the PopPUNK analysis, clade II appears to be observed more commonly in later years.

Tempest analysis of the phylogeny produced showed a negative correlation coefficient and therefore this dataset was deemed unsuitable for BEAST analysis and temporal analysis was not performed. 

### 3.4. Antigen Variants and AMR

Tetracycline resistance through the presence of the *tet32* gene was indicated in some clade II isolates. This gene was present in all isolates that were designated cluster 3 by PopPUNK and two singletons (28 and 46) but in no other groups. *TetM* gene presence was observed in 13 isolates (clade II = 5, external = 8) and these are also linked to PopPUNK cluster numbers (14 = 3, 15 = 2, 21 = 2, 23 = 2, 29 = 1, 33 = 1, 43 = 1, 58 = 1). Those within clade II (cluster 14 and 15) that have the *tetM* gene are also predicted to be resistant to macrolides (*erm*) and chloramphenicol (*cat*). Inspection of the annotated GFF files for the clade II isolates showing this resistance pattern suggested that cluster 14 isolates may contain a transposon due to the presence of a Tn3 family transposase. Isolates within cluster 15 contain similar genes within the immediate vicinity of the resistance genes but the Tn3 family transposon was not annotated in these cases. 

The three clades showed differing antigenic profiles to both each other and the external isolates (Figure 4). The majority of clade Ia showed variant type I neuraminidase (*nanA*), whereas, clades Ib and II showed variant type III and a large proportion of the external clade showed variant type II. ATP-dependent Clp protease *SP2194* showed little variation and most isolates were variant type I except for clade Ib which contained a variant type II gene. β-N-acetylhexosaminidase *strH* was variable across the clades and variant type II was seen more commonly than type I in clades Ia and II. However, in both clade Ib and the external group, variant type I was seen more commonly, with all of clade Ib containing this variant. Both clades Ia and II contained family 1 *pspA* sequences but the two clades showed higher similarity to different clusters within that family. Some clade II isolates and all clade Ib contained sequences more closely related to family 2. External isolates contained both family 1 and family 2 sequences. *PspA* sequences in some isolates could not be detected. *PspC* sequences were largely the same within clades but all three clades showed sequences from different groups (clade Ia = group 8,4; clade Ib = 11,8,6,10; clade II = 6,2,8). Other protein antigens investigated are not shown in Figure 4 as they did not show any variation among the clades or were absent in most isolates.

### 3.5. Capsular Operon Comparisons

Comparisons of capsular operon sequences showed a small amount of variation across the dataset (average distance 8.45 SNPs), with a larger SNP distance between isolates within different clades (max 91 SNPs). Many operon sequences are identical by SNP analysis. A variant in the *pgm* gene is common to all isolates within clades Ia, Ib and II. This variant results in a non-synonymous change in this gene: A207S (10038G > T in reference). A second variant, P106S in the *galU* gene (8818C > T in reference), is common to all clade II isolates. 

## 4. Discussion

Data shown here describe changes in the genomic content of serotype 3 invasive isolates in England and Wales, consistent with that seen in previous studies. All CC180 isolates fall within the clades previously described [6,7] in carriage and invasive isolates in the UK and other countries. Previous studies have excluded non-CC180 isolates; however, our dataset shows that CC505 is also within the lineages defined previously and should be considered within this context. Other sequence types observed within the time frame investigated are external to the previously described clades and represent a small proportion of the overall IPD isolates, confirming that CC180 is the major clade associated with serotype 3. The carriage isolates included in this study, although small in number, also reflect this switch, reflecting the findings of previous carriage studies [8].

The data contained within this study represent 10% of the serotype 3 isolates referred to Public Health England between 2003 and 2019. The selection process applied to the referred isolates was designed to maximise the diversity within the dataset and reduce the risk of artificially identifying similarities between isolates. Therefore, the proportions observed within this dataset are expected to be representative of the population. 

PopPUNK analysis showed good correlation between the isolates from Azarian et al. [7] and the data from this study. There are few clusters that fall within one of the three clades phylogenetically that do not contain at least one isolate from the Azarian et al. [7] dataset. Several singletons or small clusters (<4 isolates) were identified and a number of these are within the CC180 clades. It is likely that if the dataset were increased to include all serotype 3 isolates, there would be a larger number of similar isolates and fewer outliers. This is shown by the Global Pneumococcal Sequencing Cluster (GPSC) dataset [30], which groups all CC180 serotype 3 isolates into a single macro cluster because of the larger number of isolates and increased distances between clusters. Therefore, a comparison of isolates to the global dataset does not provide any greater resolution than standard MLST. This macro-clustering from global datasets highlights the benefits of the local clustering methods performed here to produce greater resolution between isolates. One complication associated with the use of kmer-based clustering to determine lineage is that new isolates that do not belong to an existing cluster, or “outliers”, and are difficult to categorise without producing a phylogeny. One potential method that will alleviate these problems could be the development of a core genome MLST (cgMLST) scheme, as this will allow measurable distances to be applied to these outliers and provide greater discrimination than conventional MLST. 

PopPUNK cluster analysis showed that the two previously defined clades Ia and II were further separated into two major groups and several smaller clusters or singletons. Within the phylogeny, the major clusters within clade Ia appear to separate into two clonal branches; however, there seems to be a greater level of mixing among the two clusters in clade II. The reference genome used for phylogenetic analysis was a clade Ia genome and therefore the clade Ia isolates will show the greatest resolution. The mixing of clusters within clade II may also result from the differences detected by PopPUNK occurring within the accessory genome which is additional to the clade Ia reference genome used. 

Clade II was unseen in this dataset before 2008 and, by 2018, represents 50% of the isolates investigated. This clade has increased in number since 2010 but the rate of change has become more apparent since 2014, at which point overall levels of serotype 3 also begin to increase rapidly. The study by Azarian et al. [7] showed that there were no significant differences in the clade II zeta potential or capsule shedding and that clade II was more likely to be susceptible to antisera than clade Ia, therefore, suggesting this increase is not directly due to evasion of the vaccine. 

In this study, we were able to identify two variants within the capsular operon able to distinguish the three clades from external isolates and distinguish clade II from clade I. As these are non-synonymous changes, they may impact the capsular operon. No other widespread variation was observed among this dataset and, in general, changes across the operon were minimal. It may be pertinent to further investigate the impact of these distinguishing variants to establish any impact they may have on the resultant polysaccharide and therefore the efficacy of the vaccine. 

A second hypothesis proposed by Azarian et al. [7] is that clade II shows an alternate antigenic and antimicrobial resistance profile, and this allows the isolates to better evade the immune system. The analysis of 13 antigenic markers showed that the clades do contain differing antigenic profiles. *PspA* shows a high level of variability and required a larger number of sequences and lower stringency mapping to identify the family present in most isolates. Clade Ib shows a very divergent antigen profile compared to clades Ia and II. This may also reflect the fact that all but two of these isolates are CC505 and therefore a level of divergence would be expected between these groups and this clonal complex may not show as much variation as observed within CC180. The two CC180 isolates within clade Ib show a different antigen profile to the CC505 clade Ib isolates, as well as the CC180 clade Ia isolates. 

In a study that categorized pneumococcal protein antigens within carriage isolates [31], *pspA* family 1 and *nanA* variant III were shown to be the most commonly observed within the pneumococcal population. These data show that *PspA* family 2 is more common among serotype 3 isolates and this was also reflected in a previous serotype 3-specific study [7]. Unlike *pspA*, the previous serotype 3 study showed that serotype 3 isolates generally contained variant I *nanA*; however, the PHE data show that clade Ia contains variant I but clades Ib and II contain genes more closely related to variant III, reported as most commonly observed in pneumococcal carriage. 

The differences in antigen profile between the clades supports the hypothesis that clade II has emerged in recent years as it is able to more successfully evade the host immune system. However, this analysis uses genomic data alone, and so these differences would have to be confirmed experimentally in any future work.

Some clade II isolates, including all isolates from one PopPUNK cluster and two singletons, are predicted to be resistant to tetracycline due to the presence of the *tet32* gene. The fact that these isolates are within a specific cluster suggests that they may have a transposable or mobile genetic element that separates them from the other major cluster, or there is a further divergence within clade II that is not evident due to the phylogeny issues discussed above. The *tetM* gene is also observed in this dataset, including some clade II isolates, some of which also contain other resistance markers. The presence of a transposable element in these isolates is suggested by the presence of a Tn3 family transposase in some isolates. Interestingly, these isolates do form a distinct subclade within clade II suggesting greater genetic diversity within these isolates. This study does not include any phenotypic confirmation of the markers identified (e.g., tetracycline resistance) and the predictions made are based on genomic data alone. However, work by Metcalf et al. [26] showed that WGS-based resistance prediction is a good alternative to phenotypic methods. 

Although this study contains approximately 10% of serotype 3 isolates since 2003, the proportion of IPD cases that were referred to PHE for serotyping was lower in earlier years. Therefore, the isolates available from early in the study period will slightly underrepresent the population of serotype 3 at that time in England and Wales. However, for recent years, isolate referral rates were higher and stringency regarding invasive isolates was higher. Therefore, the data available for analysis represents a significant proportion of the IPD in England and Wales at that time. Metadata associated with the isolates included in the study have not been taken into consideration (e.g., age, sex, etc.) and this may have biased the dataset towards a particular demographic, such as those over 65 years old.

This study has been able to describe the changes in genomic content of serotype 3 in England and Wales over an extended period, spanning pre- and post-vaccine periods. Analysis has shown that there are several differences between the observed clades within serotype 3 that could explain the recent emergence of clade II. The investigation of these differing traits in combination with extended clinical information in the future will allow a deeper understanding of these changes in a clinical context. Continued observation of the distribution of clades is essential and this study has highlighted the benefits of routine whole genome sequencing and the detailed characterization of strains. 

## Figures and Tables

**Figure 1 genes-10-00845-f001:**
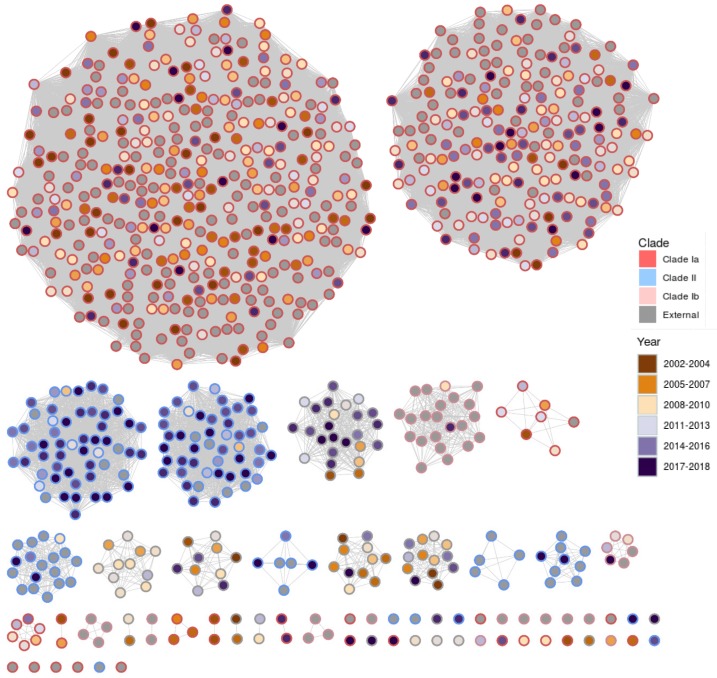
Clusters generated by PopPUNK including Public Health England (PHE) genomes and data from Azarian et al. [7]. Nodes are coloured by the year in which the sample was received by PHE. Azarian et al. [7] data points are filled in grey. The outer circle shows the clade number (grey indicates no clade was assigned).

**Figure 2 genes-10-00845-f002:**
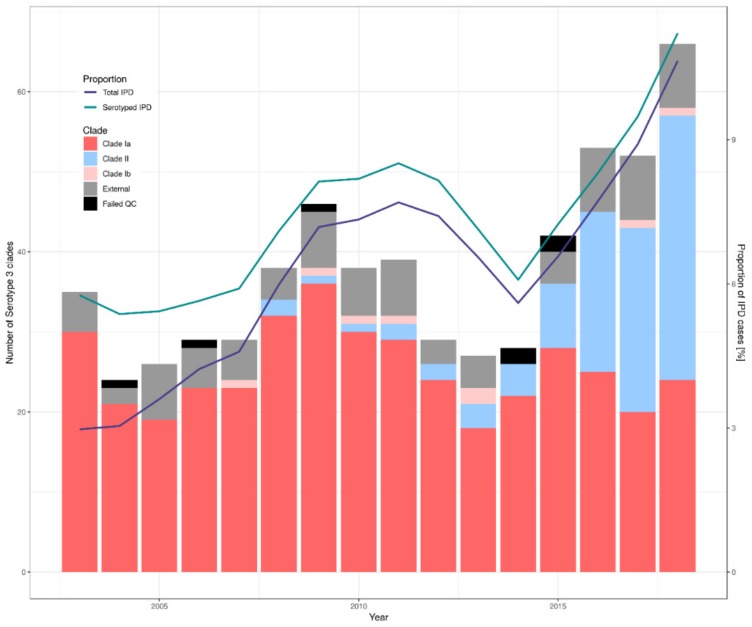
The bars show the number of *S. pneumoniae* serotype 3 isolates within each clade. The lines show the proportion of the recorded invasive pneumococcal disease (IPD) cases caused by serotype 3. The proportion of all IPD cases (purple) and of those that were serotyped (green) are shown for comparison. N.B. the failed isolates are those that did not pass the QC thresholds applied to the PHEnix pipeline. The counts included in the graph contain only isolates referred to the reference laboratory; carriage isolates were excluded.

**Figure 3 genes-10-00845-f003:**
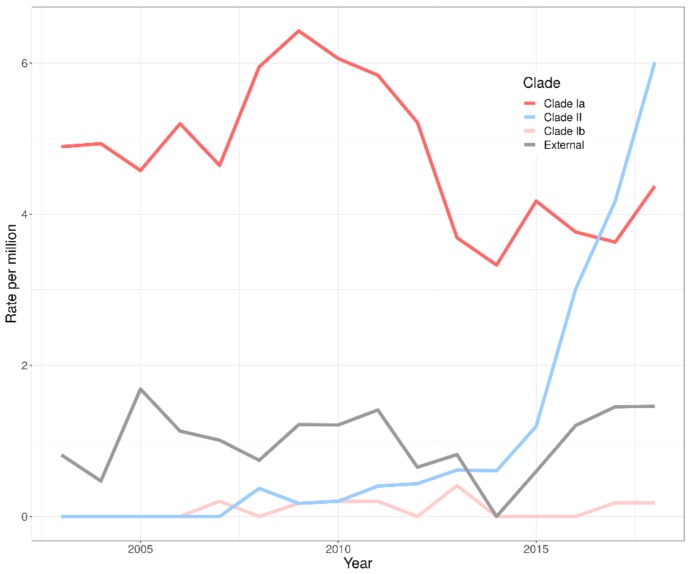
Time trend analysis for the *S. pneumoniae* serotype 3 clade groups over time. The rate is estimated per million people and shows a significant increase in clade II from 2013 and a slight decrease in clade Ia over the period.

**Figure 4 genes-10-00845-f004:**
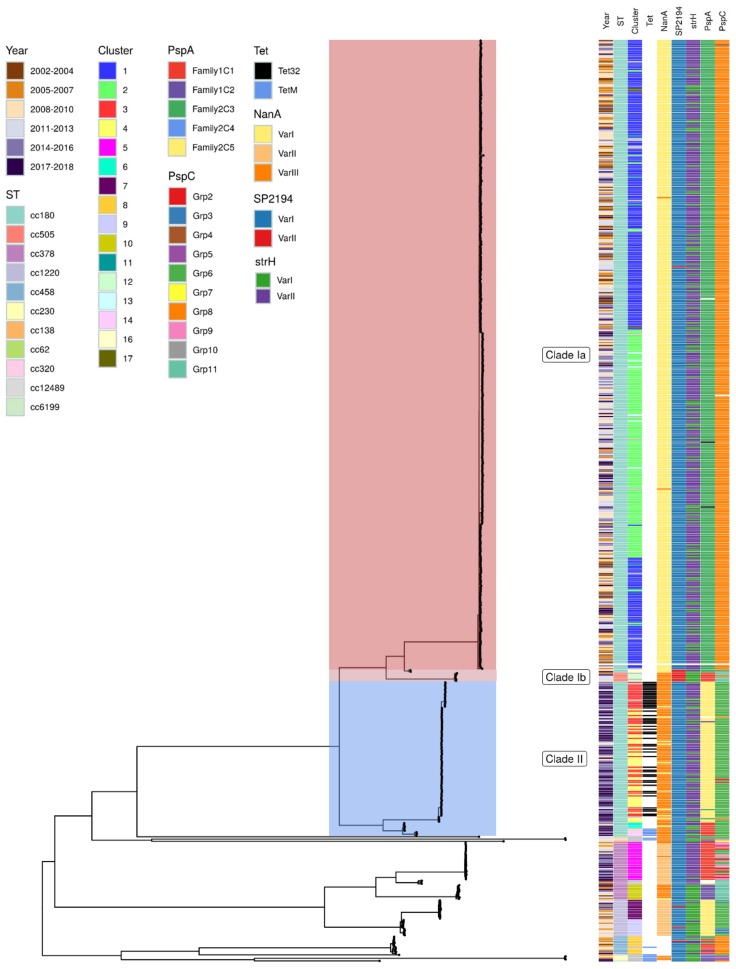
Maximum likelihood tree of all *S. pneumoniae* PHE-serotyped genomes that met the minimum QC requirements (see methods). The tree is midpoint rooted. Clades Ia (red), Ib (light pink) and II (blue) are indicated on the corresponding clades and labelled. Isolates that are not highlighted are referred to as “external”. The matrix to the right of the tree shows associated metadata including the year of sampling, sequence type (ST), PopPUNK cluster, tetracycline resistance markers, and relevant protein antigen variants. Undetected genes are shown in white and protein antigen results that did meet required QC parameters are shown in black.

**Table 1 genes-10-00845-t001:** Distribution of *S. pneumoniae* serotype 3 isolates per year including the number of non-invasive isolates from the routine isolates and the number of available carriage isolates.

Year	Number of Isolates Referred to Reference Laboratory	Routine Isolates Sequenced	% of Total	Number of Non-Invasive Isolates	Carriage Isolates	Failed PHEnix Quality Control
2002	-	0	-		3	
2003	43 *	35	-	5		
2004	227	24	10.57	6		1
2005	257	26	10.12	2		
2006	282	29	10.28	2		1
2007	292	29	9.93			
2008	379	38	10.03			
2009	452	47	10.40	2	6	1
2010	384	38	9.90	2		
2011	388	39	10.05	1		
2012	288	29	10.07	1		
2013	288	27	9.38	3	1	
2014	279	28	10.04			2
2015	425	42	9.88	3		2
2016	547	53	9.69		3	
2017	626	52	8.31			
2018	583	66	11.32		1	

* As the laboratory information management system was not introduced until 2003, total numbers are not available for 2002 and the number of isolates within 2003 does not reflect the total for that year.

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
