# Peer review of "Evolution of Streptococcus pneumoniae Serotype 3 in England and Wales: A Major Vaccine Evader"

_genes, 2019, doi:10.3390/genes10110845_

Round 1

Reviewer 1 Report

This study describes extensive whole genome sequencing of the clinically important serotype 3 of Streptococccus pneumonie in England and Wales. Genomic analysis clearly suggested that a specific phylogenic clade has, clade II has exapanded since 2014. Furthermore the authors did analysis of resistance genes and genes encoding protein antigens in the isolates.

The study is potentially clinically important and highlight the in depth information about emerging clades that appears under vaccination pressure. However, in its present form the study cannot be fully evaluated and has some problems with presentation and over-interpretation of results.

Major concerns

Figure 3 is missing, making it difficult to evalute if the interpretations are valid. There is not experimental support for that clade II has emerged because of an increaed ability to evade the host immune system. Genotype should be discussed as such. Assumptions about phenotype and immune evasion could be made, but without any analysis of transcriptome, proteome, ultrastructure or direct assays of immune evasion this is purely speculation.

Minor concerns

1. Streptococcus pneumonie should be mentioned in the title.

2. Lines 51 and 52 correct "clade II ..... than clade II"

3. References should be proofread:

a) No upper case in article titles

b) Journal names should be abbreviated consistently

c) Species should be in italics

Author Response

The authors thank the reviewer for their comments on the paper. It is unfortunate that figure 3 was not available for review in this case. It was included in the manuscript when submitted, hopefully this will not be an issue going forward. Please note that in the revised manuscript the tree is now figure 4.

The points regarding no experimental evidence supporting the increased ability to evade the immune system are well made and the authors agree with this. The wording around this hypothesis has been changed within the manuscript to better reflect the fact that these are assumptions regarding evasion and would require experimental support.

Minor concerns:

Streptococcus pneumonie should be mentioned in the title.

            - Agreed, the title has been modified accordingly

Lines 51 and 52 correct "clade II ..... than clade II"

            - Agreed, changed in manuscript

References should be proofread: a) No upper case in article titles b) Journal names should be abbreviated consistently c) Species should be in italics

             - These have been changed where required.

Reviewer 2 Report

This manuscript by Groves et al. is described for the clade distribution of invasive serotype 3 pnenumococci in England and Wales. This manuscript is written very well and the result is very clear, and they showed that clade II has expanded since 2014 and now represents half of serotype 3 IPD isolates. The clade II spreading is already published by Azarian et al. in 2018 by using global isolates. Therefore, this manuscript described the local change of clade II isolates. Therefore, this reviewer feel this manuscript needs some minor changes.

1. Title. As described above, this manuscript showed the local changes of clade II isolates in England and Wales. However, from this title, this reviewer can not read the pnemococci and area. Therefore, it is better to change the title including the pneumonocci and its localization.

2.  L. 113. The PHEnix pipeline is very useful for this analysis. However, as described in L. 115 ‘where the proportion of N bases or gaps exceeded 0.5 were removed”. N gaps are removed from VCF files. How about the phages or IS elements?

3. Lane 167. Fig. 1 Red circle for Clade I1 is hardly understand, this color seems brown.

4. L. 206 AMR genes. The existence of AMR genes. Did the authors checked the MIC of tetracyclins, macrolides or chloramphenicol? In addition, the AMR against penicillin has changed in clade II isolates? This reviewer feels the MIC date is very important to understand the spreading of specific clones such as clade II. Is it impossible to put these information?

Author Response

The authors thank the reviewer for their comments on this manuscript. We address the points raised individually below.

Reviewers comment:

Title. As described above, this manuscript showed the local changes of clade II isolates in England and Wales. However, from this title, this reviewer can not read the pnemococci and area. Therefore, it is better to change the title including the pneumonocci and its localization.

Authors response:

Agreed, the title of the manuscript has been changed accordingly

Reviewers comment:

L. 113. The PHEnix pipeline is very useful for this analysis. However, as described in L. 115 ‘where the proportion of N bases or gaps exceeded 0.5 were removed”. N gaps are removed from VCF files. How about the phages or IS elements?

Authors response:

Phages and IS elements in this case were not removed in order to allow assessment of these areas between clades should they be significant. Positions containing a high proportion of gaps or Ns were removed to reduce the impact of poorly sequenced regions of the genome and the authors felt this was sufficient for the analysis to be carried out.

Reviewers comment:

Lane 167. Fig. 1 Red circle for Clade I1 is hardly understand, this color seems brown.

Authors response:

The figure has been altered to increase the thickness of the outer rings to improve the comprehension of this figure.

Reviewers comment:

L. 206 AMR genes. The existence of AMR genes. Did the authors checked the MIC of tetracyclins, macrolides or chloramphenicol? In addition, the AMR against penicillin has changed in clade II isolates? This reviewer feels the MIC date is very important to understand the spreading of specific clones such as clade II. Is it impossible to put these information?

Authors response:

Unfortunately, we do not have MIC data for these isolates as the project did not allow for this additional work to be carried out. The authors feel that there is prior work demonstrating that resistance genes are a good predictor of phenotype. This reference has been added to the manuscript to support the use of genomics without MIC values https://www.ncbi.nlm.nih.gov/pubmed/?term=27542334.